# Mechano-dependent signaling by Latrophilin/CIRL quenches cAMP in proprioceptive neurons

Nicole Scholz[1,2†], Chonglin Guan[1†], Matthias Nieberler[1†],
Alexander Grotemeyer[1†], Isabella Maiellaro[3,4], Shiqiang Gao[5], Sebastian Beck[5],
Matthias Pawlak[1], Markus Sauer[6], Esther Asan[7], Sven Rothemund[8], Jana Winkler[9],
Simone Prömel[9], Georg Nagel[5], Tobias Langenhan[1,2*], Robert J Kittel[1*]

[1]Department of Neurophysiology, Institute of Physiology, University of Würzburg, Würzburg, Germany; [2]Rudolf Schönheimer Institute of Biochemistry, Division of General Biochemistry, Medical Faculty, Leipzig University, Leipzig, Germany; [3]Institute of Pharmacology and Toxicology, University of Würzburg, Würzburg, Germany; [4]Rudolf Virchow Center, DFG-Research Center for Experimental Biomedicine, University of Würzburg, Würzburg, Germany; [5]Department of Biology, Institute for Molecular Plant Physiology and Biophysics, University of Würzburg Biocenter, Würzburg, Germany; [6]Department of Biotechnology and Biophysics, University of Würzburg Biocenter, Würzburg, Germany; [7]Institute of Anatomy and Cell Biology, University of Würzburg, Würzburg, Germany; [8]Core Unit Peptide Technologies, Medical Faculty, Leipzig University, Leipzig, Germany; [9]Rudolf Schönheimer Institute of Biochemistry, Division of Molecular Biochemistry, Medical Faculty, Leipzig University, Leipzig, Germany

*For correspondence: tobias. langenhan@gmail.com (TL); robert.kittel@uni-wuerzburg.de (RJK)

†These authors contributed equally to this work

Competing interests: The authors declare that no competing interests exist.

**Abstract** Adhesion-type G protein-coupled receptors (aGPCRs), a large molecule family with over 30 members in humans, operate in organ development, brain function and govern immunological responses. Correspondingly, this receptor family is linked to a multitude of diverse human diseases. aGPCRs have been suggested to possess mechanosensory properties, though their mechanism of action is fully unknown. Here we show that the *Drosophila* aGPCR Latrophilin/ dCIRL acts in mechanosensory neurons by modulating ionotropic receptor currents, the initiating step of cellular mechanosensation. This process depends on the length of the extended ectodomain and the tethered agonist of the receptor, but not on its autoproteolysis, a characteristic biochemical feature of the aGPCR family. Intracellularly, dCIRL quenches cAMP levels upon mechanical activation thereby specifically increasing the mechanosensitivity of neurons. These results provide direct evidence that the aGPCR dCIRL acts as a molecular sensor and signal transducer that detects and converts mechanical stimuli into a metabotropic response.

## Introduction

Sensory strategies for the perception of mechanical cues are essential for survival. However, our understanding of the underlying molecular mechanisms is far from complete. G protein-coupled receptors (GPCRs) hand over stimulus-induced conformational changes to metabotropic signaling outlets that carry the signal to intracellular destinations.

Adhesion-type G protein-coupled receptors (aGPCRs) display structural characteristics that distinguish them as a separate family within the GPCR superfamily (*Hamann et al., 2015*). Remarkably, as

opposed to the majority of GPCRs, aGPCRs interact through their N-termini with membrane-tethered or ECM-fixed partner molecules rather than soluble compounds indicating that their function requires positional fixation outside the receptor-bearing cell (*Langenhan et al., 2013*).

Several aGPCRs have recently been linked to mechanosensitive functions (*Petersen et al., 2015*; *Scholz et al., 2015*; *White et al., 2014*). These examples collectively suggest that processing of mechanical stimuli may be a common feature of this receptor family (*Langenhan et al., 2016*). However, while elemental signaling properties of aGPCRs have recently become available (*Hamann et al., 2015*), a molecular model of their signal transduction strategy is at large.

By combining genomic engineering with electrophysiological recordings, super-resolution microscopy and optogenetics, we have determined the critical steps that are required to transduce a mechanical stimulus into an intracellular response by an individual aGPCR, *Drosophila* Latrophilin/ dCIRL. We have taken advantage of the functional modulation of mechanosensory neurons by dCIRL and the accessibility of this system for physiological interrogation in vivo. Our results show that dCIRL is located in the neuronal dendrites and cilia of chordotonal organs (ChOs), the sites of ionotropic mechanotransduction (*Ranade et al., 2015*). dCIRL specifically shapes the generation of mechanically-gated receptor currents but is dispensable for normal membrane excitability of ChO neurons. Lengthening dCIRL's N-terminal fragment (NTF) gradually reduces mechanosensory neuronal responses. This is consistent with a model in which mechanical tension applied to the receptor determines the extent of its activity. In contrast, autoproteolysis of the GAIN domain is not essential for dCIRL activity, which instead requires an intact *Stachel* sequence. Finally, we show that mechanical stimuli effect a dCIRL-dependent decrease of cAMP levels in ChO neurons.

## Results

### dCIRL is located in dendrites and cilia of mechanosensory neurons

To precisely determine the expression of *dCirl* in larval mechanosensory chordotonal organs (ChOs), we used a *dCirlp^GAL4* promoter element to drive the nuclear reporter *UAS-GFP::nls* and analyzed immunohistochemical stainings against GFP and HRP, a comarker of ChO neuron structure. In the larval pentascolopidial ChO (lch5) only the five neuronal nuclei were marked (*Figure 1a*), showing that *dCirl* is a neuronal gene. To obtain a translational expression profile of dCIRL, we constructed a genomic transgene that contains an mRFP cassette inserted into an exon encoding part of the extracellular domain (ECD) of the receptor at a position where its folding and trafficking should not be affected (*dCirl^N-RFP*; *Figure 1—figure supplement 1*) (*Scholz et al., 2015*). The dCIRL^N-RFP fusion protein could be observed in the lch5 at the level of the dendrite and cilia (*Figure 1b*). Next, we employed super-resolution imaging by structured illumination microscopy (SIM) to resolve the subcellular arrangement of dCIRL in greater detail (*Gustafsson, 2000*). SIM images depicted a patchy distribution of dCIRL^N-RFP at the membrane of the lch5 dendrite and cilium, where it localized near the TRP channel TRPN1/NompC (*Yan et al., 2013*; *Zhang et al., 2015*) (*Figure 1c*). This demonstrates that dCIRL resides at the location where ionotropic mechanosensation operates.

### The ultrastructure of *dCirl^KO* chordotonal organs is unaffected

As dCIRL possesses molecular characteristics of adhesion molecules, we performed ultrastructural analyses to ascertain that removal of *dCirl* does not affect the complex architecture and structural integrity of ChOs. Scanning electron microscopy uncovered no structural anomalies in *dCirl^KO* mutants (*dCirl^Rescue*: n = 11 ChOs from 5 larvae; *dCirl^KO*: n = 11 from 5 larvae; *Figure 1d,e*). Additionally, the ultrastructure, cell-cell and cell-matrix contacts of distal inner dendrites and cilia appeared unaltered in transmission electron microscopy (*dCirl^Rescue*: n = 9 ChOs from 6 larvae; *dCirl^KO*: n = 15 ChOs from 8 larvae; *Figure 1—figure supplement 2*). Thus, all ChO cell types and their serial interconnections are present in the mutant demonstrating that removal of *dCirl* does not interrupt the complex architecture and cytology of the larval lch5. This corroborates earlier findings, based on fluorescence microscopy of molecular markers (*Scholz et al., 2015*), that *dCirl* is not involved in the structural specialization of ChOs.

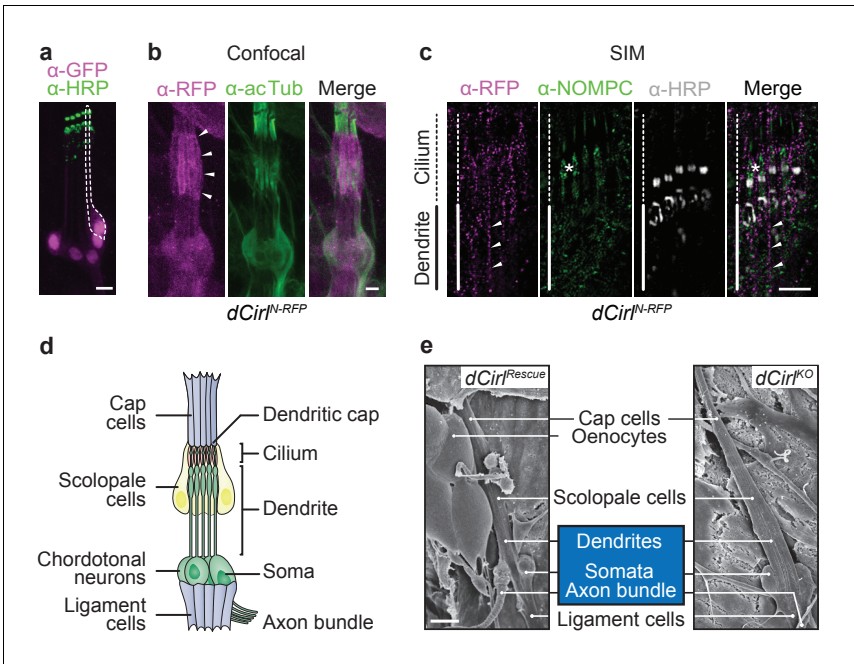

**Figure 1.** dCIRL is located at the site of ionotropic mechanosensation. (**a**) The *dCirlp^GAL4* driver demonstrates exclusively neuronal expression of *dCirl* within lch5 ChOs (*dCirlp^GAL4>UAS-GFP::nls*). Rightmost mechanosensory neuron (soma and dendrite) within the organ marked by a dotted line. (**b**) Maximal projection of a confocal image stack of lch5 (counterstained against acetylated tubulin; green) showing dCIRL^N-RFP (magenta) enrichment at the level of the distal dendrites and cilia (arrowheads). (**c**) SIM imaging shows *dCirl^N-RFP* (magenta) in the distal dendrites (arrowheads) extending to the ciliary compartment, where the receptor is coexpressed in the same subcellular region with the TRP channel NompC (green). lch5 was counterstained with α-HRP, asterisk indicates ciliary dilation. Note that SIM resolves the canal through which the cilium passes. (**d**) Composition of the larval pentascolopidial organ (lch5). (**e**) Scanning electron micrographs of lch5 from control and *dCirl^KO* animals. The organ consists of a chain of support cell types that suspend the mechanosensory neurons (blue) between body wall and musculature. No morphological abnormalities are apparent in the mutant. Scale bars, (**a–c**) 5 µm; (**e**) 10 µm. See also *Figure 1—figure supplements 1* and *2*.

The following figure supplements are available for figure 1:

**Figure supplement 1.** *dCirl* genomic engineering platform.

**Figure supplement 2.** Transmission electron microscopy of ChO in control and *dCirl^KO*.

## Optogenetic stimulation of chordotonal neurons bypasses dCIRL-dependence

Two qualitatively different forms of electrical activity mediate signal transduction and transformation in primary sensory neurons, such as the bipolar nerve cells of ChOs. During transduction, stimulus encounter by sensory receptors is converted into current flow through ion channels to generate the receptor potential. This membrane depolarization is then transformed into a train of action potentials by voltage-gated ion channels to carry the sensory signal along the axon. dCIRL increases the mechanically-induced firing frequency of ChO neurons (*Scholz et al., 2015*). We reasoned that the light-gated cation channel Channelrhodopsin-2 (*Nagel et al., 2003*) [ChR2; retinal-bound channelopsin-2 (Chop2)] could be used to distinguish whether this effect was exerted at the level of mechanosensory transduction or transformation. Because ChOs are also thermoresponsive (*Liu et al., 2003*), this strategy necessitated an efficient ChR variant to limit the heat generated by the required light intensities. We therefore screened for a ChR2 version that combines high photostimulation efficiency (*Dawydow et al., 2014*) with good temporal precision. The D156H mutant displayed very high expression in *Xenopus* oocytes upon inspection by confocal microscopy (*Figure 2a*), while retaining

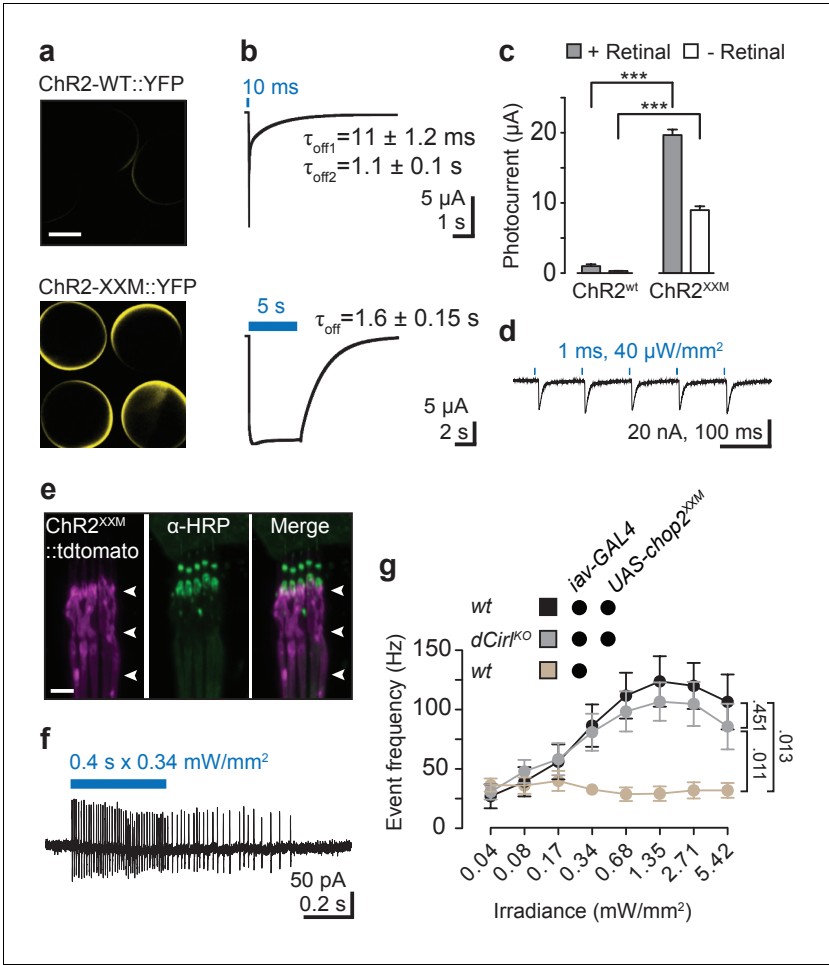

**Figure 2.** Optogenetic stimulation with ChR2-XXM. (**a**) Expression of ChR2-WT::YFP and ChR2-XXM::YFP in *Xenopus* oocytes (without retinal supplementation) imaged by confocal microscopy. (**b**) Representative photocurrents of ChR2-XXM::YFP in oocytes (473 nm, ~12.4 mW/mm$^2$). Short light pulses are followed by a rapid biphasic photocurrent decay ($\tau_{off1}$: 80%, $\tau_{off2}$: 20%), whereas the longer time constant ($\tau_{off}$) dominates upon prolonged photostimulation. Data are presented as mean ± SD, n = 4 recordings from individual oocytes incubated with 1 µM all-*trans*-retinal. (**c**) Quantification of photocurrent amplitudes in oocytes with and without retinal supplementation. Data presented as mean ± SEM. ChR2-wt + retinal: 0.999 ± 0.5272 µA, n = 4; ChR2-wt - retinal: 0.317 ± 0.0570 µA, n = 5; ChR2-XXM + retinal: 19.675 ± 1.9458 µA n = 6; ChR2-XXM - retinal: 8.982 ± 1.5718 µA, n = 8; p<0.00001, Student's *t*- test. (**d**) Two-electrode voltage clamp (TEVC) recordings at the NMJ show that photostimulation of motoneurons (440 nm) via ChR2-XXM::tdTomato elicits excitatory postsynaptic currents (EPSCs), which can be stimulus-locked using short, low intensity light pulses. (**e**) Localization of ChR2-XXM:: tdTomato in lch5 dendrites (arrowheads). (**f**) Example recording from the lch5 axon bundle showing a train of action currents elicited by photostimulation of sensory neurons via ChR2-XXM::tdTomato. The burst gradually decays after the light pulse, reflecting the kinetics of channel closure. (**g**) Quantification of action current frequencies in lch5 neurons expressing ChR2-XXM::tdTomato upon increasing irradiance. The activity of ChOs scales with light intensity and is independent of *dCirl*. No light response when the transgene is omitted. Data are presented as mean ± SEM. n = 10 per genotype. Numbers denote *p* values of comparisons of event frequency at 5.42 mW/mm$^2$ irradiance with a Student's *t*- test. Scale bars, (**a**) 500 µm; (**e**) 5 µm. See also *Figure 2—figure supplements 1* and *2*.

The following figure supplements are available for figure 2:

**Figure supplement 1.** Characterization of ChR2-XXM at the NMJ.

**Figure supplement 2.** Stimulation of larval ChO neurons via ChR2-XXM in vivo.

favorable kinetic properties, especially after short light pulses (10 ms: $\tau_{off1}$ = 11 ± 1.2 ms SD, $\tau_{off2}$ = 1.1 ± 0.13 s SD; *Figure 2b*), and over ten-fold larger photocurrents than the wildtype version (ChR2-wt; *Figure 2c*). We therefore named the ChR2$^{D156H}$ variant ChR2-XXM (e<u>x</u>tra high e<u>x</u>pression and <u>m</u>edium open state).

Imaging, electrophysiological recordings and in vivo assays confirmed the utility of ChR2-XXM at the neuromuscular junction (NMJ; *ok6-GAL4*; *Figure 2d*, *Figure 2—figure supplement 1*) and in ChO neurons (*iav-GAL4*; *Figure 2e,f*, *Figure 2—figure supplement 2*) of *Drosophila*. To examine whether *dCirl* supports the initiation of action potentials in mechanosensory neurons, we recorded from the Ich5 axon bundle during photostimulation via ChR2-XXM. Photoinduced action current frequencies were indistinguishable in control and *dCirl$^{KO}$* animals over the entire irradiance spectrum (*Figure 2g*). Thus, by bypassing the receptor potential, this optogenetic approach demonstrates that dCIRL does not promote membrane excitability per se to help initiate and propagate action potentials in the sensory neuron.

## Chordotonal organs sense temperature changes independently of dCIRL

Because ChOs respond to temperature changes (*Liu et al., 2003*) we tested whether dCIRL also processes this non-mechanical stimulus. Action current frequencies in lch5 afferents gradually increased with rising temperature, roughly doubling from 15°C to 30°C (*Figure 3a,b*). Notably, *dCirl$^{KO}$* neurons displayed unaltered thermosensory electrical activity, while bouts of mechanical vibration evoked lower action current frequencies in the mutant. Interestingly, this difference was most pronounced at

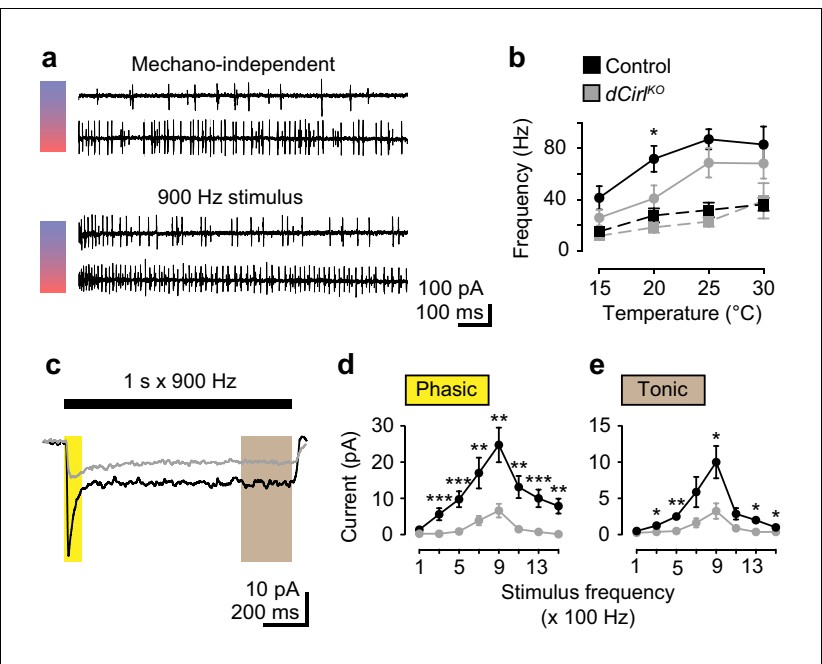

**Figure 3.** dCIRL shapes mechanosensory signal transduction. (**a**) Recordings of wildtype lch5 action currents at 15°C and 30°C without and during mechanical vibration at 900 Hz applied to the cap cell. (**b**) Quantification of action current frequencies without (dashed line) and with (solid line) mechanical stimulation in control (black) and *dCirl$^{KO}$* larvae (gray). Asterisk denotes p≤0.05 comparing event frequency at 20°C with a Student's *t*-test. Data are presented as mean ± SEM, n = 8 animals per genotype. (**c**) Current recordings from lch5 neurons during 900 Hz mechanical stimulation in the presence of TTX (average of 10 sweeps). The wildtype (black) receptor current displays phasic (yellow shaded area) and tonic (gray area) components, both of which are strongly reduced after removal of *dCirl* (gray). (**d**) Quantification of phasic and (**e**) tonic current amplitudes across a stimulation range from 100 to 1500 Hz. Data are presented as mean ± SEM, n = 8 per genotype. Asterisks denote comparisons of current amplitude with a Mann-Whitney U test (*p≤0.05, **p≤0.01).

20°C and was partially compensated by low and high temperatures (*Figure 3b*). These findings demonstrate that dCIRL plays a mechano-specific role in this sensory organ.

## dCIRL increases mechanically triggered receptor currents

Next, we blocked voltage-gated sodium channels with tetrodotoxin (TTX) to isolate mechanosensory receptor currents. As a result, the initiation of action potentials is prevented and isolated receptor currents can be assessed. Both phasic and tonic current components were strongly reduced in $dCirl^{KO}$ neurons (*Figure 3c–e*), providing direct evidence that dCIRL modulates the receptor potential evoked by mechanical stimulation.

We observed that a diminished yet graded receptor current profile persisted upon increasing vibrational cues even in the absence of *dCirl*. This feature further attests to the fact that dCIRL controls the sensitivity of mechanosensory neurons towards mechanostimulation rather than the neurons' principal ability to respond to mechanical challenge.

## dCIRL NTF length determines mechanosensitivity of chordotonal neurons

Characteristic of aGPCRs, dCIRL possesses a long extracellular N-terminus with adhesive properties that anchors the receptor to the extracellular matrix or to opposed cell surfaces via cognate ligands. By applying mechanical tension to the ECD this setting may facilitate the reliable transmission of mechanical deformation to the receptor. We sought to test this hypothesis by relaxing dCIRL's extracellular region via gradual elongation of the ECD through the insertion of spacer elements. All transgenic constructs were expressed from the genomic *dCirl* locus (*Figure 1—figure supplement 1*) (*Scholz et al., 2015*) and a small Bungarotoxin binding site fused to a hemagglutinin tag ($dCirl^{BBS::HA}$) served as an insertion site control. Action current frequencies of $dCirl^{BBS::HA}$ neurons were comparable to wildtype indicating that cassette insertion did not interfere with structure or expression of the receptor (*Figure 4a,b*). Elongating the ECD through an mRFP cassette ($dCirl^{N-RFP}$), which adds at least 2 nm, blunted the response at 900 Hz and a substantial length increase by the 3xCD4 spacer marked with poly-V5 tags ($dCirl^{3xCD4}$; *Figure 4a,c*), which adds approximately 20 nm, flattened the activity profile across the entire stimulation range (*Figure 4b*). We therefore hypothesize that ECD length and tensile properties may adjust dCIRL's response towards mechanical challenge (*Figure 4d*).

## Autoproteolytic processing is dispensable for dCIRL activity

All aGPCRs contain a juxtamembrane GPCR autoproteolysis inducing (GAIN) domain (*Araç et al., 2012*), which catalyzes receptor cleavage in N and C-terminal fragments (NTF, CTF) and maintains the two non-covalently affixed (*Gray et al., 1996*). This unusual property may be required for protein folding and trafficking (*Prömel et al., 2013*) or to expose the receptor's tethered agonist (*Stachel*), which begins at the GPCR proteolysis site (GPS; *Figure 5a*) (*Krasnoperov et al., 1997*; *Lin et al., 2004*) and can potently stimulate receptor activity (*Liebscher et al., 2014*; *Stoveken et al., 2015*). To test this assumption, we abolished autoproteolytic activity of the GAIN domain in two sets of *dCirl* alleles by mutating the $-2$ ($dCirl^{H>A}$) or $+1$ ($dCirl^{T>A}$) position of the GPS ($H^{-2}L^{-1}{\downarrow}T^{+1}$; *Figure 5a,b*) (*Prömel et al., 2012*), notably the latter within the *Stachel* sequence. In the first set, the GPS mutations were inserted into the RFP-tagged receptor background ($dCirl^{N-RFP/H>A}$, $dCirl^{N-RFP/T>A}$), and in the second set, the unmodified *dCirl* template was mutated ($dCirl^{H>A}$, $dCirl^{T>A}$). We prepared protein extracts from $dCirl^{N-RFP/H>A}$ and $dCirl^{N-RFP/T>A}$ flies and immunoblotted against the RFP tag. Both mutant proteins were detected as a full-length band of ca. 218 kDa (*Figure 5b*). In contrast, the 106 kDa band, which corresponds to the RFP-tagged dCIRL NTF, was not present (*Figure 5b*). This shows that both GPS mutations abrogated the autoproteolytic activity of the dCIRL GAIN domain.

SIM images of immunostained mechanosensory neurons revealed that autoproteolysis is not required for membrane targeting of dCIRL to dendritic and ciliary compartments (*Figure 5c*). Interestingly, however, mechanically-induced receptor currents (*Figure 5d,e*) were differently affected by the two mutations. Whereas $dCirl^{H>A}$ neurons displayed wildtype responses, the $dCirl^{T>A}$ mutant delivered a null phenotype. These results demonstrate that dCIRL activation in vivo depends on an intact tethered agonist, but that NTF-CTF disruption is dispensable.

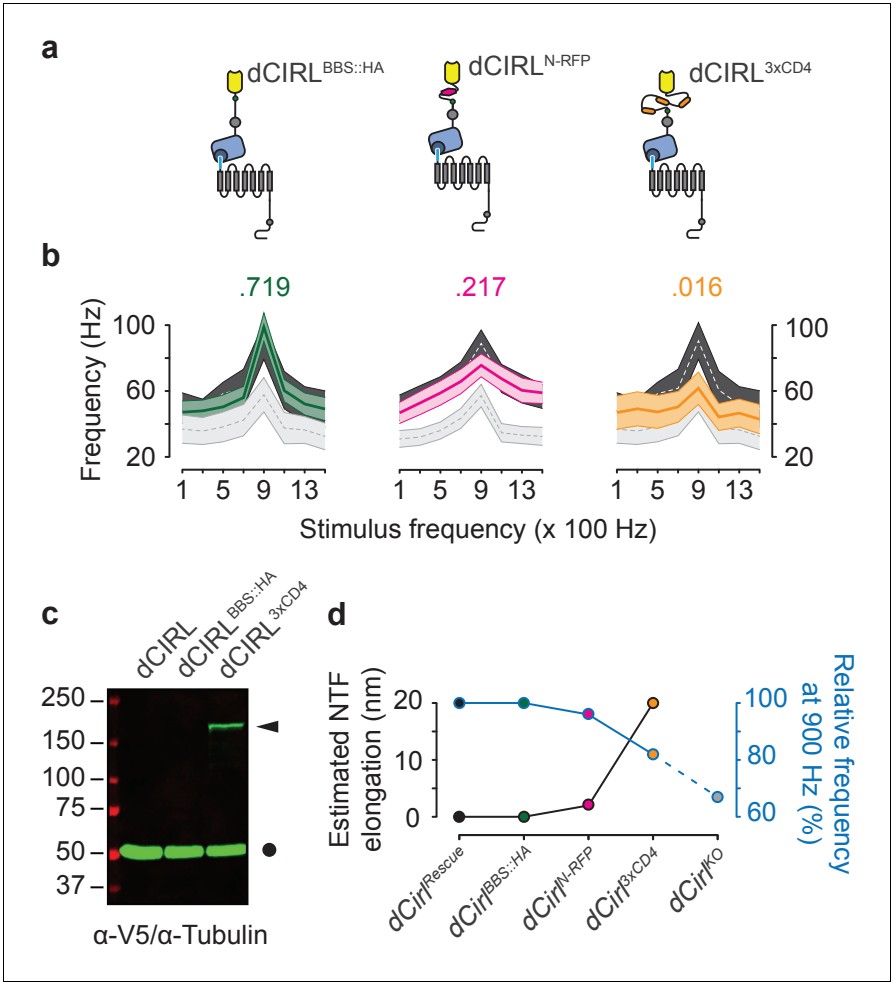

**Figure 4.** Extending the dCIRL NTF reduces the mechanosensory response. (**a**) Upper panel, protein design of dCIRL elongation constructs bestowed with an HA::BBX fusion tag (left, green circle), an mRFP moiety (middle, magenta hexagon), or a triple CD4 immunoglobulin repeat cassette (right, orange ovals). All spacers were integrated into the same site within the dCIRL NTF just C-terminal of the RBL (rhamnose-binding lectin) domain. Schematics not to scale. (**b**) Action current frequencies plotted against mechanical stimulation. Response curves of wildtype (*dCirl^Rescue*; dark gray) and knockout (*dCirl^KO*; light gray) lch5 neurons recorded in the same experiment are displayed for comparison. Data are presented as mean ± SEM. *dCirl^BBS::HA*/*dCirl^Rescue*/*dCirl^KO* (n = 10/20/20); *dCirl^N-RFP*/*dCirl^Rescue*/*dCirl^KO* (n = 20/20/20); *dCirl^3xCD4*/*dCirl^Rescue*/*dCirl^KO* (n = 10/20/20). Numbers above plots denote *p* values of comparisons with a Student's *t*-test between *dCirl^Rescue* and respective elongated *dCirl* variants at 900 Hz stimulation, n denotes number of larvae. (**c**) Western blot showing stable expression of the dCIRL^3xCD4 fusion protein in vivo. Protein extracts from animals (10 per genotype) were blotted and immunostained with an α-V5 antiserum specifically detecting the elongated NTF of dCIRL^3xCD4 (ca. 177 kDa) bestowed with poly-V5-tags (arrowhead). Consistent with previous results on the high efficiency of GAIN-mediated dCIRL autoproteolysis (*Scholz et al., 2015*), no full-length receptor was found. α-Tubulin staining was used as loading control (circle). (**d**) Relationship between estimated NTF elongation (black curve) and lch5 response frequency (blue curve), normalized to respective *dCirl^Rescue* responses.

## Mechanostimulation of dCIRL decreases the cAMP concentration in mechanosensory neurons

To interrogate intracellular signaling by dCIRL we chose an optogenetic approach by utilizing the photoactivated adenylyl cyclase bPAC (*Stierl et al., 2011*) (*iav-GAL4>UAS-bPAC*). Photoinduced cAMP elevation in wildtype lch5 quenched neuronal activity to the level observed in *dCirl^KO* mutants, while bPAC activation in the *dCirl^KO* background did not further decrease action current frequencies

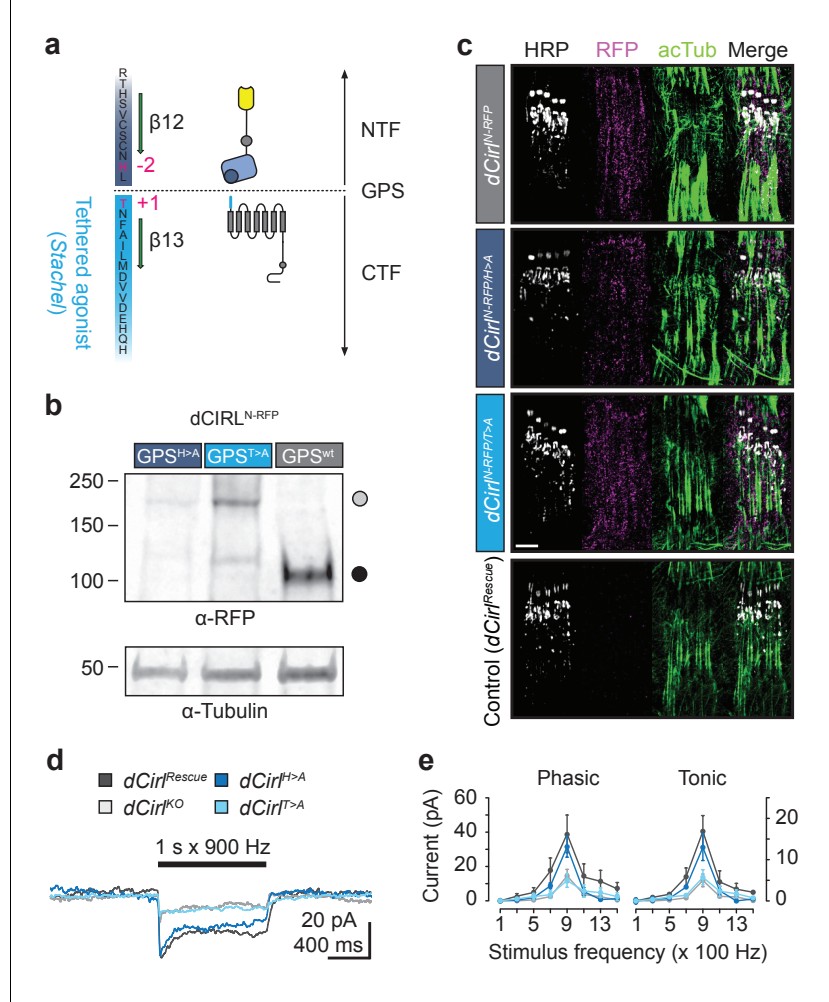

**Figure 5.** Differential effect of GPS mutations on mechanosensitivity. (**a**) Structure of the dCIRL GPS region. The GPS separates NTF from CTF in proteolyzable aGPCRs. The C-terminal cleavage component contains the *Stachel* sequence, a potent receptor agonist in many aGPCRs (light blue). Magenta: conserved, mutated residues that are necessary for GPS cleavage. (**b**) Western blot of whole fly protein extracts containing wildtype or proteolysis-defective GPS variants of dCIRL probed against an mRFP tag in the NTF. The dCIRL-GPS$^{wt}$ sample displays only a fragment corresponding to the cleaved NTF (ca. 106 kDa; filled circle), while the two GPS mutants contain a band representing the full-length receptor (ca. 218 kDa; open circle). (**c**) SIM images of dCIRL$^{N-RFP}$ fusion proteins with wildtype and proteolysis-resistant GPS in lch5. The protein is trafficked into dendrites and cilia, regardless of autoproteolytic cleavage. Scale bar 5 μm. (**d**) Receptor current recordings (average of 8 sweeps) of lch5 neurons under TTX inhibition highlight the divergent effects of the GPS mutations on mechanosensitivity (dark blue, *dCirl*$^{H>A}$; light blue, *dCirl*$^{T>A}$). (**e**) Quantification of tonic and phasic receptor current components. Despite abrogating GPS cleavage, the response profile of the *dCirl*$^{H>A}$ receptor variant is unaffected (900 Hz, phasic: p=0.464, tonic: p=0.460, Student's *t*-test vs. *dCirl*$^{Rescue}$). In contrast, changing the first residue of the *Stachel* sequence in *dCirl*$^{T>A}$ mutants abolishes the receptor's mechanosensory function, resulting in a *dCirl*$^{KO}$ response profile (900 Hz, phasic: p=0.030, tonic: p=0.023, Student's *t*-test vs. *dCirl*$^{Rescue}$). Data are presented as mean ± SEM, n = 8 larvae per genotype.

significantly (*Figure 6a–c*). Conversely, pharmacological inhibition of adenylyl cyclase activity specifically rescued *dCirl*$^{KO}$ neuron function (*Figure 6d*). These observations indicate that increased cAMP levels attenuate the mechanosensory response and suggest that dCIRL modulates neuronal activity by suppressing cAMP production.

Next, we employed the FRET-based cAMP sensor Epac1-camps (*Maiellaro et al., 2016*; *Nikolaev et al., 2004*) to directly visualize neuronal cAMP dynamics during mechanical stimulation

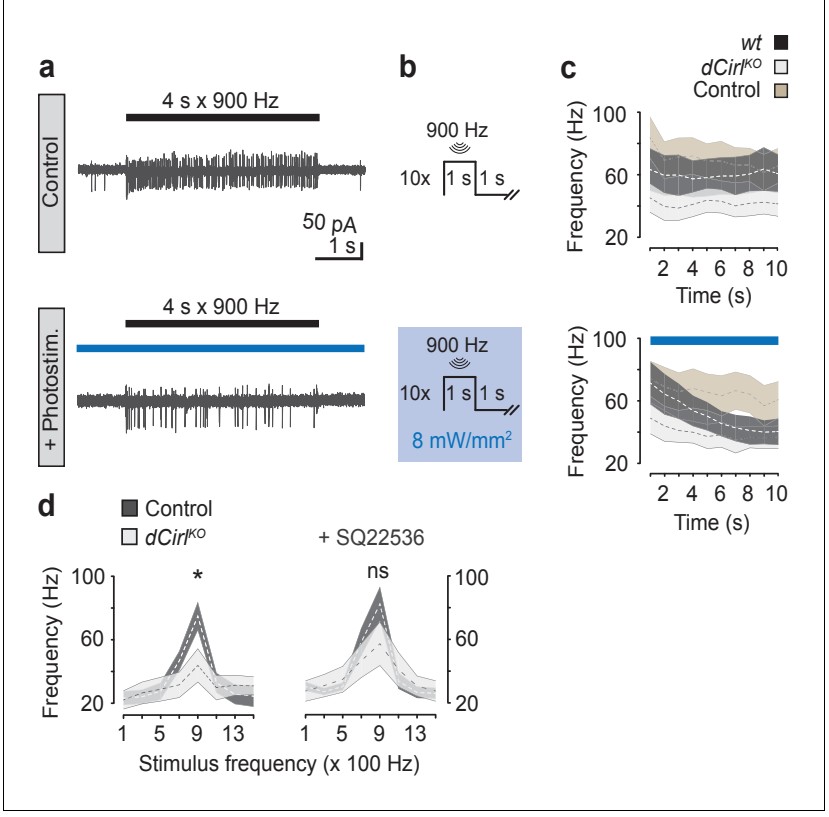

**Figure 6.** cAMP signaling by dCIRL. (a) Example current recordings from wildtype lch5 neurons during only mechanical (upper panel) and combined mechanical-light stimulation (lower panel) demonstrate the suppressive effect of cAMP elevation by bPAC on the mechanically-evoked action current frequency. (b) Protocol for combined mechanical stimulation and optogenetic cAMP production via bPAC photoactivation. (c) The mechanosensory response (action current frequency) of wildtype lch5 neurons is decreased to the level of *dCirl^{KO}* larvae by increasing cAMP concentrations through light-induced bPAC stimulation (blue bar). In contrast, *dCirl^{KO}* neurons are unaffected by light stimulation. Data are presented as mean ± SEM, n denotes number of animals. *iav-GAL4>UAS-bPAC; wt* (black, n = 9); *iav-GAL4>UAS-bPAC; dCirl^{KO}* (gray, n = 10); *iav-GAL4; wt* (brown, n = 9). (d) Pharmacological inhibition of adenylyl cyclase activity using 100 µM SQ22536 rescues mechanically-evoked action current frequencies in *dCirl^{KO}* lch5 neurons. Data are presented as mean ± SEM. Event frequency at 900 Hz without inhibitor: Control: 74.9 ± 8.67 Hz; *dCirl^{KO}*: 43.88 ± 10.48 Hz; p=0.0287, Student's *t*-test. Event frequency at 900 Hz with inhibitor: Control: 82.63 ± 10.51 Hz; *dCirl^{KO}*: 57.25 ± 13.69 Hz; p=0.2103; n = 8 per genotype and condition.

(*Figure 7a*). Application of the adenylyl cyclase agonist forskolin (FSK) produced similar relative FRET changes in wildtype and *dCirl^{KO}* neurons, indicating comparable basal cAMP levels (*Figure 7—figure supplement 1*). However, whereas bouts of mechanical vibration reproducibly triggered a cAMP decrease in wildtype neurons, this second messenger signal was abrogated in *dCirl^{KO}* mutants (*Figure 7b,c*). This was corroborated by coupling assays of dCIRL, in which a 12 amino acid synthetic peptide (P12), corresponding to the receptor's *Stachel* sequence, was sufficient to stimulate Gα_i (*Figure 7—figure supplement 2*).

## Discussion

Here we demonstrate how a GPCR can specifically shape mechanotransduction in a sensory neuron in vivo. This study thus serves a two-fold purpose. It delineates pivotal steps in the activation paradigm of aGPCRs and sheds light on the contribution of metabotropic signals to the physiology of neuronal mechanosensation.

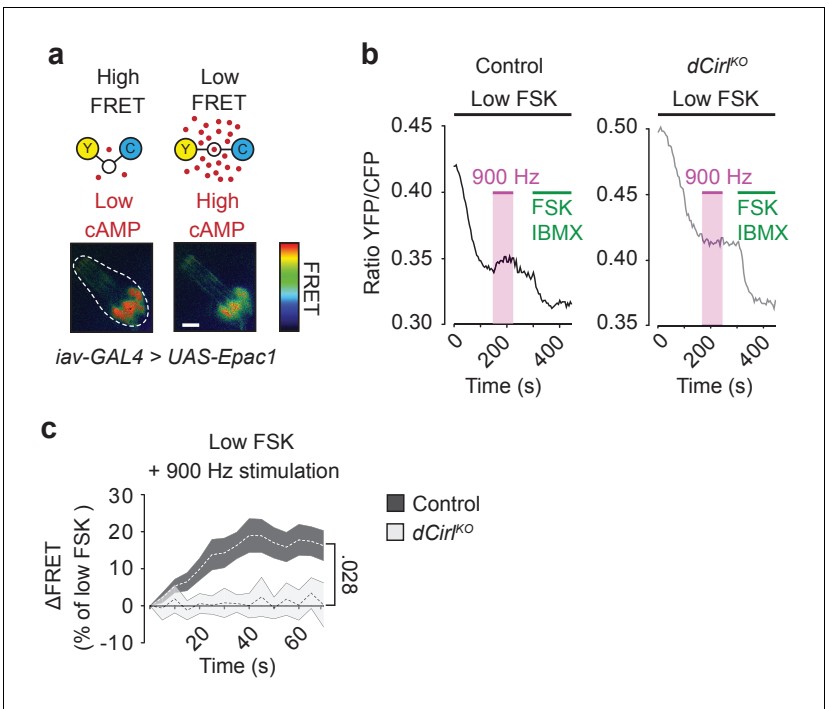

**Figure 7.** dCIRL reduces cAMP levels in sensory neurons in response to mechanical stimulation. (a) Schematic structure of the cAMP sensor Epac1-camps, which changes its conformation and fluorescence property upon binding of cAMP. Corresponding pseudocolor FRET images (YFP/CFP ratios) of Ich5 neurons (*iav-GAL4>UAS-Epac1-camps*) at low and high cAMP concentrations. Scale bar 10 μm. (b) Absolute FRET values (YFP/CFP ratios) recorded in control and *dCirl^KO* Ich5 neurons, corresponding to the region of interest depicted in (a). In order to ensure a dynamic sensor range, 0.5 μM FSK was first added to the preparation (*Maiellaro et al., 2016*). Mechanical stimulation (900 Hz, pink bar) decreases cAMP levels in control but not in *dCirl^KO* Ich5 neurons. At the end of the experiment, maximal FRET responses are induced by 10 μM FSK and 100 μM IBMX (3-Isobutyl-1-methylxanthin), a non-selective phosphodiesterase inhibitor. (c) Average time course of piezo-induced FRET changes in control and *dCirl^KO* Ich5 neurons. Data are expressed as percentages of the low forskolin response and presented as mean ± SEM. ΔFRET at 70 s: Control: 16.28 ± 4.05%, n = 14; *dCirl^KO*: 0.147 ± 3.78%, n = 6 larvae. Number denotes $p$ value of comparison at 70 s with a Student's $t$-test. See also *Figure 7—figure supplements 1* and *2*.

The following figure supplements are available for figure 7:

**Figure supplement 1.** Basal cAMP levels in ChO neurons.

**Figure supplement 2.** A synthetic peptide mimicking dCIRL's tethered agonist stimulates Gα_i coupling.

While there is ongoing discussion whether metabotropic pathways are suitable to sense physical or chemical stimuli with fast onset kinetics, due to the supposed inherent slowness of second messenger systems (*Knecht et al., 2015*; *Wilson, 2013*), our results demonstrate that the aGPCR dCIRL/Latrophilin is necessary for faithful mechanostimulus detection in the lch5 organ of *Drosophila* larvae. Here, dCIRL contributes to the correct setting of the neuron's mechanically-evoked receptor potential. This is in line with the location of the receptor, which is present in the dendritic membrane and the single cilium of ChO neurons, one of the few documentations of the subcellular location of an aGPCR in its natural environment. The dendritic and ciliary membranes harbor mechanosensitive Transient Receptor Potential (TRP) channels that elicit a receptor potential in the mechanosensory neuron by converting mechanical strain into ion flux (*Cheng et al., 2010*; *Kim et al., 2003*; *Zhang et al., 2015*). Moreover, two mechanosensitive TRP channel subunits, *TRPN1/NompC* and *TRPV/Nanchung*, interact genetically with *dCirl* (*Scholz et al., 2015*). The present study further

specifies this relationship by showing that the extent of the mechanosensory receptor current is controlled by *dCirl*. This suggests that the activity of the aGPCR directly modulates ion flux through TRP channels, and highlights that metabotropic and ionotropic signals may cooperate during the rapid sensory processes that underlie primary mechanosensation.

The nature of this cooperation is yet unclear. Second messenger signals may alter force-response properties of ion channels through post-translational modifications to correct for the mechanical setting of sensory structures, e.g. stretch, shape or osmotic state of the neuron, before acute mechanical stimuli arrive. Indeed, there are precedents for such a direct interplay between GPCRs and channel proteins in olfactory (*Connelly et al., 2015*) and cardiovascular contexts (*Chachisvilis et al., 2006*; *Mederos y Schnitzler et al., 2011*; *2008*; *Zou et al., 2004*).

ChOs are polymodal sensors that can also detect thermal stimuli (*Liu et al., 2003*). We show that dCIRL does not influence this thermosensory response (between 15°C and 30°C) emphasizing the mechano-specific role of this aGPCR. Replacing sensory input by optogenetic stimulation supports this conclusion, as ChR2-XXM evoked normal activity in *dCirl^{KO}* larvae.

Turning to the molecular mechanisms of dCIRL activation, we show that the length of the extracellular tail instructs receptor activity. This observation is compatible with an extracellular engagement of the dCIRL NTF with cellular or matricellular protein(s) through its adhesion domains. Mammalian latrophilins were shown to interact with teneurins (*Silva et al., 2011*), FLRTs (*O'Sullivan et al., 2014*) and neurexins 1$\beta$ and 2$\beta$ (*Boucard et al., 2012*) suggesting that the receptors are anchored to opposed cell surfaces through their ligands. However, FLRTs do not exist in *Drosophila* and an engagement of dCIRL with the other two candidate partners could not be detected to date (N.S. and T.L., unpublished observations) indicating that other interactors may engage and mechanically affix dCIRL. Our data support a model where the distance between ligand-receptor contact site and signaling 7TM unit determines the mechanical load onto the receptor protein and its subsequent signal output. This scenario bears similarity to the role of the cytoplasmic ankyrin repeats of NompC, which provide a mechanical tether to the cytoskeleton of mechanosensory cells, and are essential for proper mechanoactivation of this ionotropic sensor (*Zhang et al., 2015*).

aGPCR activation occurs by means of a tethered agonist (*Stachel*) (*Liebscher et al., 2014*; *Monk et al., 2015*; *Stoveken et al., 2015*), which encompasses the last $\beta$-strand of the GAIN domain. Structural concerns imply that after GAIN domain cleavage a substantial part of the *Stachel* remains enclosed within the GAIN domain and should thus be inaccessible to interactions with the 7TM domain (*Araç et al., 2012*; *Prömel et al., 2013*). These considerations beg the question how the tethered agonist gets exposed to stimulate receptor activity, and how this process relates to the mechanosensitivity of aGPCRs. Two models account for the elusive link between these critical features (*Langenhan et al., 2013*; *Liebscher et al., 2013*). Mechanical challenge to the receptor causes: (1) physical disruption of the heterodimer at the GPS thereby exposing the tethered agonist. In this scenario, GPS cleavage is absolutely essential for receptor activity; (2) Allosteric changes of the GAIN domain, e.g. through isomerization of the tethered agonist-7TM region, that allow for the engagement of the *Stachel* with the 7TM. In this situation, GPS cleavage and disruption of the NTF-CTF receptor heterodimer are not necessary for receptor activity. We found that autoproteolytic cleavage is not required for the perception and transduction of vibrational mechanical stimuli by dCIRL.

We further uncovered that the concomitant disruption of *Stachel* and autoproteolysis disables dCIRL's mechanosensory function in ChO neurons. Thus, the tethered agonist concept (*Monk et al., 2015*) pertains to aGPCRs in *Drosophila*. Notably, these findings also demonstrate that classical GPS mutations have similar biochemical but different physiological effects in vivo.

Finally, we interrogated intracellular signaling by dCIRL. In contrast to previously described G$\alpha_s$ coupling of rat and nematode latrophilins (*Müller et al., 2015*), the mechanosensory response of ChO neurons was decreased by optogenetic augmentation of adenylyl cylcase activity, and the mechanosensory deficit of *dCirl^{KO}* mutants was rescued by pharmacological inhibition of adenylyl cyclase. FRET measurements also directly demonstrated that mechanical stimulation reduces the cAMP concentration in the sensory neurons, and that this mechano-metabotropic coupling depends on dCIRL. Thus, dCIRL converts a mechanosensory signal into a drop of cAMP levels. This suggests that the *Drosophila* latrophilin entertains a cascade that inhibits adenylyl cyclases or stimulates

phosphodiesterases in ChO neurons, and that G-protein coupling pathways by latrophilin homologs may depend on species and/or cell type.

Members of the aGPCR family are associated with a vast range of physiological processes extending beyond canonical neuronal mechanosensation. For example, dysfunction of ADGRG1/GPR56 causes polymicrogyria (*Piao et al., 2004*), ADGRF5/GPR116 controls pulmonary surfactant production (*Bridges et al., 2013*), genetic lesions in many aGPCR loci are associated with a roster of cancer types (*Kan et al., 2010*; *O'Hayre et al., 2013*) and ADGRE2/EMR2 regulates mast cell degranulation (*Boyden et al., 2016*). Intriguingly, a point mutation in the GAIN domain of ADGRE2 sensitizes the receptor to mechanical stimuli in kindreds of patients suffering from vibratory urticaria. Our results now provide a basis to test the generality of the concept that aGPCRs are metabotropic mechanosensors also outside classical mechanosensory structures, and aid in understanding the contribution of ailing aGPCR signaling in diseased tissues.

## Materials and methods

### Fly culture conditions and stocks

Flies were raised at 25°C on standard cornmeal and molasses medium. The following strains were generated in this study:

LAT159, $w^{1118}$; $dCirl^{KO}$ {$w^{+mC}$=pMN4[$dCirl^{N-RFP}$]}attP$^{dCirl}$loxP/CyoGFP w-;; ($dCirl^{N-RFP}$)
LAT163, $w^{1118}$; $dCirl^{KO}$ {$w^{+mC}$=pTL370[$dCirl^{Rescue}$]}attP$^{dCirl}$loxP/CyoGFP w-;; ($dCirl^{Rescue}$)
LAT174, $w^{1118}$; $dCirl^{KO}$ {$w^{+mC}$=pMN9[$dCirl^{T>A}$]}attP$^{dCirl}$loxP/CyoGFP w-;; ($dCirl^{T>A}$)
LAT176, $w^{1118}$; $dCirl^{KO}$ {$w^{+mC}$=pMN10[$dCirl^{N-RFP/T>A}$]}attP$^{dCirl}$loxP/CyoGFP w-;; ($dCirl^{N-RFP/T>A}$)
LAT206, $w^{1118}$; $dCirl^{KO}$ {$w^{+mC}$=pNH98[$dCirl^{3xCD4}$]}attP$^{dCirl}$loxP/CyoGFP w-;; ($dCirl^{3xCD4}$)
LAT207, $w^{1118}$; $dCirl^{KO}$ {$w^{+mC}$=pTL564[$dCirl^{BBS::HA}$]}attP$^{dCirl}$loxP/CyoGFP w-;; ($dCirl^{BBS::HA}$)
LAT280, $w^{1118}$; $dCirl^{KO}$ {$w^{+mC}$=pMN44[$dCirl^{H>A}$]}attP$^{dCirl}$loxP/CyoGFP w-;; ($dCirl^{H>A}$)
LAT282, $w^{1118}$; $dCirl^{KO}$ {$w^{+mC}$=pMN38[$dCirl^{N-RFP/H>A}$]}attP$^{dCirl}$loxP/CyoGFP w-;; ($dCirl^{N-RFP/H>A}$)
RJK258, $w^{1118}$; {$w^{+mC}$=pTL538[chop2-D156H(XXM)]}attP$^{VK00018}$/Cyo;; (chop2$^{XXM}$)
RJK300, $w^{1118}$; {$w^{+mC}$=pTL537[chop2-D156H(XXM)::tdtomato]}attP$^{VK00018}$/CyoGFP w-;; (chop2$^{XXM}$::tdtomato)

The following strains were previously generated:
$w^{1118}$; $dCirl^{KO}$;; (*Scholz et al., 2015*)
$w^{1118}$;; P{$w^{+mC}$=iav-GAL4}attP$^2$;; (*Scholz et al., 2015*)
$w^{1118}$; $dCirlp^{GAL4}$;; (*Scholz et al., 2015*)
$w^{1118}$;; P{$w^{+mC}$=UAS-GFP::nls}8; (BDSC#4776)
w*; ok6-GAL4;; (*Sanyal, 2009*)
w*; UAS-bPAC/CyO;; (*Stierl et al., 2011*)
$w^{1118}$;; UAS-Epac1-camps $w^+$/Sb (*Maiellaro et al., 2016*)

### Transgene construction

*pMN4:* A 0.1 kb fragment annealed from primers *mn_1F/2R* containing a 3x*flag*-tag flanked by two *Age*I sites was inserted into the genomic *dCirl* construct *pTL393* (*Scholz et al., 2015*) at its *Nco*I site. Subsequently, a 0.7 kb fragment including a monomeric RFP cassette was amplified from *pTL391* using primers *mn_3F/4R* and introduced in the resulting clone via *Age*I in order to replace the *3xFlag-tag* sequence.

*pMN9:* T>A GPS cleavage-deficient *dCirl* was created with QuikChange site-directed mutagenesis of *pTL370* using primers *mn_12F/13R* containing the altered GPS sequence.

*pMN9:* T>A GPS cleavage-deficient *dCirl* was created with QuikChange site-directed mutagenesis of *pTL370* using primers *mn_12F/13R* containing the altered GPS sequence.

*pMN10:* T>A GPS cleavage-deficient *dCirl*$^{N-RFP}$ containing the extracellular mRFP cassette was created with QuikChange site-directed mutagenesis of *pMN4* using primers *mn_12F/13R* containing the altered GPS sequence.

*pMN38:* H>A GPS cleavage-deficient *dCirl*$^{N-RFP}$ containing the extracellular mRFP cassette was created with QuikChange site-directed mutagenesis of *pMN4* using primers *mn_38F/39R* containing the altered GPS sequence.

*pMN44:* H>A GPS cleavage-deficient *dCirl* was created with QuikChange site-directed mutagenesis of *pTL370* using primers *mn_38F/39R* containing the altered GPS sequence.

*pNH98:* The 3xCD4 coding region interspersed each with six *V5-tags* was engineered from MWG Eurofins (*pNH95*). Subsequently, a 2.8 kb *Age*I fragment of *pNH95* was cloned into *pMN4*.

*pTL512:* The cDNA of the *dCirl* E splice variant was amplified from EST clone *RE25258* obtained from the Drosophila Genomics Resource Center using primers *tl_508F/509R* and cloned into *pCR-BluntII-TOPO* (Thermo Fisher Scientific). A 150 bp fragment encoding the signal peptide of human *GPR56* and a *HA-tag* was amplified with primers *tl_514F/515R* from a template vector and inserted into the plasmid via *Apa*I/*Eco*RV generating *pTL506*. A 5.1 kb *Bgl*II/*Spe*I fragment was released from *pTL506* and inserted into the *pcDps* backbone generating *pTL512*.

*pTL518:* A 0.2 kb fragment was amplified off *pTL370* (*Scholz et al., 2015*) with primers *tl_540F/549R*, cut with *Eco*RV and inserted into the *Eco*RV site of *pTL506* to complete the RBL domain coding region.

*pTL520:* An annealed fragment of primers *tl_542F/543R* was ligated into the *Age*I site of *pTL512*.

*pTL521:* An annealed fragment of *tl_542F/543R* was ligated into the *Age*I site of *pTL518*.

*pTL526:* A 2.2 kb *Spe*I/*Afe*I-fragment of *pTL507* was ligated with a 6.1 kb *Spe*I/*Afe*I-fragment of *pTL520*.

*pTL535:* A 0.15 kb fragment encoding the signal peptide of the mouse ADGRL1/LPHN1 receptor was amplified off *pSP113* (*Müller et al., 2015*), cut with *Eco*RI and *Bgl*II and inserted into *pTL526*.

*pTL536:* A 2.2 kb *Spe*I/*Afe*I-fragment of *pTL507* was ligated with a 6.3 kb *Spe*I/*Afe*I-fragment of *pTL521*. A 0.15 kb fragment, amplified from *pSP113* with primers *tl_550F/551R*, was cut with *Eco*RI and *Bgl*II and inserted into the resultant plasmid.

*pTL564:* To generate the *dCirl* length sensor control construct, which includes a single Bungarotoxin binding site and hemagglutinin-tag in the RBL-HRM connecting region, a 3.5 kb *Mlu*I/*Pac*I fragment was released from *pTL555* (subclone of exons 3–6 of *dCirl* tagged with Bungarotoxin-HA-tag in *pMCS5* backbone) and inserted into *pTL393* (*attB*-flanked genomic *dCirl* wild-type construct).

*pTL665:* A 4.9 kb *Age*I/*Xba*I fragment of *pMN12* was cloned into *pTL655*.

*pTL666:* A 5.1 kb *Age*I/*Xba*I fragment of *pMN13* was cloned into *pTL655*.

*pTL696:* A 2.9 kb fragment was amplified off *pTL526* using primers *tl_727F/728R*. A second 3.4 kb fragment was amplified off *pSA3* using primers *tl_729F/696R*. Both fragments were fused using the Gibson cloning kit (NEB).

*pTL697:* A 2.9 kb fragment was amplified off *pTL526* using primers *tl_730F/728R*. A second 3.4 kb fragment was amplified off *pSA3* using primers *tl_729F/696R*. Both fragments were fused using the Gibson cloning kit (NEB).

All QuikChange-based PCRs were performed with *pfu* polymerase (Agilent). All amplicons were validated by restriction analyses followed by sequencing of the entire amplified exonic region.

Primer sequences (5'- 3'):

*mn_3F*: taaccggtgctgctgcagctgcctcctccgaggac

*mn_4R*: ataccggtagccgctgcagcggcgccggtggagtg

*mn_12F*: cagttgcaaccacctggcaaactttgccatact

*mn_13R*: agtatggcaaagtttgccaggtggttgcaactg

*mn_38F:* gcgtctgcagttgcaacgccctgacaaactttgcc

*mn_39R:* ggcaaagtttgtcagggcgttgcaactgcagacgc

*tl_508F*: atgcgatatctttcccaagtcactcagc

*tl_509R*: gtgctctagacttagccagtggttccagataacat

*tl_514F*: gtcgtagggcccactagtagatctgccaccatgactccccagtcgct

*tl_515R*: tacacggatatcaccggtggcgtagtcggggacgt

*tl_525F*: ctagacagctggattacaaggatgacgacgataagtagactagtgtcgaca

*tl_526R*: agcttgtcgacactagtctacttatcgtcgtcatccttgtaatccagctgt

*tl_540F*: agatatctccaagtaccaaaccgcctacg

*tl_542F*: ccggtgaattcaacgggaccgagggcccaaacttctacgtgcctttctccaacaagacgggcgtggtgcgca

*tl_543R*: ccggtgcgcaccacgcccgtcttgttggagaaaggcacgtagaagtttgggccctcggtcccgttgaattca

*tl_549R*: agatatcgcagttaacactccactccaca

*tl_550F*: ggaagatctgccaccatggcccgcttggctgca

*tl_551R*: cgaattcggcgtagtcggggacgtcgtaggg

*tl_727F*: actacaaaatcacccagacaaactttgccatactaatg

*tl_728R*: tcgtcatccttgtaatccttagccagtggttccag
*tl_730F*: actacaaaatcacccagttgttcaccatgttcgatggaaacat

## PhiC31-mediated recombination into *dCirl^{KO}-attP*

An established Cre recombinase-based protocol was used to remove the $w^+$-marker located in close proximity to the *dCirl* locus in *dCirl^{KO}* (*Huang et al., 2009*, *2008*). $w^+$-marked vectors bestowed with an *attB* site were injected into *phiC31[3xP3-RFP-3xP3-GFP-vas-PhiC31]; dCirl^{KO} attP-loxP;;* embryos (*Scholz et al., 2015*). Subsequently, $w^+$ served as the selection marker to identify recombinants. Precise transgene insertion was validated by PCR genotyping.

## *Xenopus* oocyte expression

cRNA was generated with the AmpliCap-MaxT7 High Yield Message Maker Kit (Epicentre Biotechnologies) using a *Nhe*I-linearized pGEM-HE XXM YFP plasmid. Oocytes were injected with 20 ng cRNA and incubated in ND96 solution (96 mM NaCl, 2 mM KCl, 1 mM $MgCl_2$, 1 mM $CaCl_2$, 10 mM HEPES and 50 µg/ml Gentamycin, pH 7.4) containing 1 µM all-*trans*-retinal (short retinal; Sigma-Aldrich), unless indicated otherwise.

## Fluorescence microscopy

### Immunohistochemistry

Staining procedures were essentially performed as previously described for the NMJ (*Ehmann et al., 2014*). For ChO imaging the following protocol was applied: third instar larvae were dissected in $Ca^{2+}$-free HL-3 (*Stewart et al., 1994*), fixed in 4% paraformaldehyde for 10 min at room temperature (RT) and blocked overnight at 4°C in 1% PBT (PBS with 1% Triton X-100, Sigma-Aldrich) supplemented with 5% normal goat serum (NGS) and 2% BSA. Primary antibodies were diluted in 1% PBT (5% NGS, 2% BSA) incubated at 4°C overnight. Next, the samples were rinsed twice and washed 3 × 20 min using 1% PBT. The secondary antibodies were added to 1% PBT (with 5% NGS) and incubated overnight at 4°C. After the samples were rinsed twice and washed 3 × 20 min with 1% PBT they were covered with Vectashield and stored for at least overnight at 4°C. The following primary antibodies were used: AcTub, 1:400 (RRID:AB_477585), ms-α-NompC (*Lee et al., 2010*) (1:200; RRID:AB_2568530), rabbit-α-GFP (1:500; RRID:AB_10790912), rabbit-α-RFP (1:500; RRID:AB_10781500). Secondary antibodies: α-HRP conjugated with Cy3 (RRID:AB_2338959) and Cy5 (*Jan and Jan, 1982*) (1:250, Dianova), α-HRP conjugated with Alexa Fluor-488 (*Jan and Jan, 1982*) (1:250; RRID:AB_2338965), Phalloidin conjugated with Alexa 488 (1:500; RRID:AB_2315147), Alexa Fluor-488-conjugated goat-α-mouse (RRID:AB_2534069) and goat-α-rabbit (RRID:AB_143165; each 1:250), Cy3-conjugated goat-α-rabbit (1:250; RRID:AB_2338006). Samples were mounted in Vectashield (Vector Laboratories). Confocal images were acquired with an LSM 5 Pascal (Zeiss) and for ChR2 stainings 100 µM retinal was added to the food.

### SIM

SIM images were recorded and processes with a commercial inverted SIM microscope (Zeiss Elyra) equipped with an oil-immersion objective (Plan-Apochromat 63x, NA 1.4 Oil Dic M27). Standard laser illumination at 488 nm, 561 nm and 642 nm was used for excitation of Alexa Fluor-488, Cy3 and Cy5-conjugated antibodies, respectively. Stacks of at least 5 planes were recorded with structured illumination from 5 rotational and 5 phase variations and processed with standard Elyra settings.

## Scanning electron microscopy

Larvae were dissected in ice-cold $Ca^{2+}$-free HL-3 and fixed overnight at RT using 6.25% glutaraldehyde in Sörensen buffer (pH 7.4; 50 mM $KH_2PO_4$, 50 mM $Na_2HPO_4$). The larval filets were washed 5 × 5 min in 100 mM Sörensen buffer and subsequently dehydrated in an aceton series (in percent: 30, 50, 75, 90, 100). Each incubation step lasted at least 30 min. Samples were transferred into teflon vessels, critically point dried (Critical Point Dryer, BAL-TEC CPD030) and adhered to 0.5 inch aluminium specimen stubs (Agar Scientific G301). Samples were placed into a Sputter Coater (BAL-TEC SCD005), flooded 3–4 times with argon *in vacuo* and subsequently metalized with gold-palladium. Imaging was done using a JEOL JSM-7500F equipped with a secondary-electron detector (SEI).

## Transmission electron microscopy

Third instar larvae were dissected in ice-cold $Ca^{2+}$-free HL3 (*Stewart et al., 1994*) and prepared for transmission electron microscopy essentially as previously described (*Wagh et al., 2006*; *Wagner et al., 2015*). Briefly, after dissection, the larval filets were fixed in 2.5% glutaraldehyde and 2.5% paraformaldehyde in either 0.1 M cacodylate buffer (CB) pH 7.3 for 2 hr at 4°C (Fix I) or in 0.05 M CB pH 7.2 for 45 min at 4°C (Fix II). For Fix I, the larvae were washed overnight in 4.5% sucrose in 0.1 M CB at 4°C, postfixed with 2% osmiumtetroxide in 0.014 M veronal acetate buffer pH 7.3 (VB, with 0.02% $CaCl_2$ and 2.25% sucrose added) for 1.5 hr, washed in VB and dehydrated in ascending concentrations of ethanol. For Fix II, all steps including dehydration (see below) were carried out at 4°C. Larvae were washed in 0.05 M CB and postfixed in 2% osmiumtetroxide in the same buffer for 1.5–2 hr followed by contrasting with 0.5% aqueous uranyl acetate (UA) overnight, washing in dH2O and dehydrating in ethanol. After dehydration, all preparations were transferred to Epon via propylene oxide as intermedium, flat embedded in Epon, ultrathin sectioned (~80 nm), and contrasted with uranyl acetate (UA) and lead citrate according to standard protocols. Ultrathin sections were analyzed using a LEO 912 AB transmission electron microscope (Zeiss). Both fixation protocols gave similar results, with slightly better ultrastructure preservation using Fix I. Digitally recorded electron micrographic images were composed and adjusted for brightness and contrast using Photoshop (Adobe).

## Immunoblots

Fly heads were collected in standard radioimmunoprecipitation assay buffer (RIPA buffer; 150 mM NaCl, 1% Triton X-100, 0.5% sodium deoxycholate, 0.1% SDS, 50 mM Tris [pH 8.0]) supplemented with protease inhibitor cocktail (1:1000; Sigma-Aldrich) and immediately frozen in liquid nitrogen. Next, heads were homogenized and supplemented with SDS-based protein buffer (Li-cor) and 2-mercaptoethanol (Merck). Next, samples were centrifuged for 5 min at 13,000 rpm (4°C), incubated for 10 min at 55°C, subjected to electrophoresis on a 4–12% Tris-Glycin SDS gel (Invitrogen) and blotted onto 0.2 µm nitrocellulose membrane (AmershamProtran). The membrane was blocked for 1 hr using Odyssey Blocking buffer (Li-cor) diluted 1:8 with 1 x PBS.

For dCIRL[3xCD4] detection ten fly heads of each genotype were collected and immediately frozen using liquid nitrogen. Subsequently, 20 µl 2% SDS was added and a glas stirrer was used to grind the heads before 8 µl of 4x Sample buffer (Li-cor) and 2 µl of 10% Triton X-100 was supplemented. Samples were cooked for 5 min at 95°C and centrifuged for 15 min at 13,000 rpm at RT. Gel electrophoresis was done using 4–12% Tris Glycine gels (Invitrogen). Protein was blotted onto 0.2 µm nitrocellulose membrane (Li-cor), blocked for 1 hr using Odyssey Blocking buffer (Li-cor) diluted 1:1 with 1x PBS.

Blots were probed with primary antisera at the indicated concentrations for 1 hr at RT: rabbit-α-RFP (1:500), mouse-α-tubulin$\beta$ (1:1000, RRID:AB_528499), mouse-α-V5 (1:500; RRID:AB_2556564). After rinsing twice and 3 × 10 min washing steps, membranes were incubated with IRDye 680RD goat-α-rabbit (RRID:AB_10956166) and 800CW goat-α-mouse (1:15000; RRID:AB_10956588) for 1 hr at RT, and again rinsed twice and washed 3 × 10 min. Western blots were imaged with an OdysseyFc 2800 (Li-cor).

## Electrophysiology

### Chordotonal neurons

Electrophysiological measurements were essentially carried out as previously described (*Scholz et al., 2015*). In brief, activity of lch5 neurons was recorded from the axon bundle using a suction electrode coupled to an EPC 10 USB amplifier (HEKA Instruments) and analyzed in Clampfit 10.2 (Molecular Devices). Mechanical stimulation was applied through a piezo-actuated, fire-sealed glass electrode placed on the muscle covering the cap cells. Spontaneously active neurons were stimulated optogenetically or at the indicated sine wave frequencies (three cycles of 1 s stimulation preceded by 1 s rest for each frequency). Data were sampled at 10 kHz and a notch filter was used to remove the specific stimulation frequency from the current trace. Pharmacological inhibition of adenylyl cyclase activity followed a full series of mechanical stimulation. Preparations were then incubated for 10 min with 100 µM SQ22536 (Merck) to inhibit adenylyl cyclase activity (*Gao and Raj, 2001*) before applying a second set of mechanical stimulation.

Light from a mercury lamp (Nikon Intensilight C-HGFI) passed a GFP filter (460–500 nm band-pass) for photostimulation of lch5 neurons via ChR2-XXM::tdTomato (*iav-Gal4>UAS-chop2^XXM::tdTomato*; 100 µM retinal food supplementation). Increasing light intensities (approx. 0.04, 0.08, 0.17, 0.34, 0.68, 1.35, 2.71, 5.42 mW/mm$^2$) were applied with intermittent 10 s breaks. For bPAC experiments (*iav-Gal4>UAS-bPAC*), first 10 cycles of 1 s mechanical stimulation at 900 Hz followed by 1 s rest were applied without irradiation. After a 3 s break, this stimulation block was paired with continuous light stimulation (460–500 nm; ~8 mW/mm$^2$).

In order to isolate receptor currents, 4 µM TTX was added to the bath to block action potentials. For each frequency, either ten (*Figure 2j–l*) or three stimulation cycles (*Figure 3g,h*) were applied (1 s stimulation preceded by 1 s rest). Traces were low-pass filtered at 30 Hz before measuring the amplitudes of phasic (peak response) and tonic current components (average of last 200 ms).

Genotypes were blinded for all electrophysiological recordings of ChOs.

### NMJ

Larvae expressing *ChR2-XXM::tdTomato* in motoneurons (*ok6-Gal4>UAS-chop2^XXM::tdTomato*) were raised in food supplemented with 100 µM retinal and dissected in ice-cold, Ca$^{2+}$-free HL-3 (in mM: NaCl 70, KCl 5, MgCl$_2$ 20, NaHCO$_3$ 10, trehalose 5, sucrose 115, HEPES 5, pH adjusted to 7.2). The VNC was removed, the peripheral nerves were severed and two-electrode voltage clamp recordings were made from ventral longitudinal muscle 6 (clamped at −60 mV) in abdominal segments A2 and A3 at room temperature, in principle as previously described (*Ljaschenko et al., 2013*). Light-evoked EPSCs were triggered by blue light (440 nm; CoolLED) in HL-3 containing 1 mM CaCl$_2$. Data were acquired with an Axoclamp 900A amplifier (Molecular Devices), signals were sampled at 10 kHz, low-pass filtered at 1 kHz and analysed with Clampfit 10.2.

### Oocytes

Two-electrode voltage-clamp recordings were performed with a conventional setup (amplifier: Turbo TEC-05 npi) at a holding potential of −100 mV in Ringer's solution (110 mM NaCl, 5 mM KCl, 2 mM BaCl$_2$, 1 mM MgCl$_2$, 5 mM HEPES, pH 7.6). Photocurrents were evoked by a water-cooled diode pumped solid-state laser (473 nm, 12.4 mW/mm$^2$). Recordings were obtained using WinEDR 3.4.2 (J. Dempster, University of Strathclyde) and stationary photocurrents were analyzed using pClamp 10.3.2 (Molecular Devices).

## Optogenetics in vivo
### Chordotonal neurons

Larvae expressing ChR2-XXM::tdTomato in mechanosensory neurons (*iav-Gal4>UAS-chop2^XXM::tdTomato*; 100 µM retinal food supplementation) were placed in a petri dish (10 cm diameter, filled with 1% agar) and recorded under infrared illumination. In each set of experiments, seven larvae were analyzed for 30 s before and during illumination with blue LEDs (440 nm, ~3 µW/mm$^2$). During light stimulation, the head swinging phase was defined as the time interval between repeated lateral movements of the anterior segment and two complete crawling sequences in forward direction.

### NMJ

Light from a mercury lamp passed through a GFP excitation band-pass filter was used to photostimulate crawling larvae expressing tagged or untagged ChR2-XXM in motoneurons (*ok6-Gal4* driver; 100 µM retinal food supplementation unless indicated otherwise). Measurements denote the time between light-induced immobilization and resumed movement (defined as anterior displacement of posterior end) during ongoing irradiation. Adult flies were transferred to a vertically positioned Petri dish (10 cm diameter) and stimulated with blue LEDs (440 nm) for 10 s. After 5 s, the dish was tapped and the immobilized individuals were counted.

## FRET-based cAMP measurements

Ratiometric FRET imaging was performed using an upright epifluorescence microscope (Axio Observer, Zeiss) equipped with a water-immersion objective (63x, NA 1.1), a xenon lamp coupled to a monochromator (VisiView, VisiChrome), filters for CFP (436/20, 455LP dichroic) and YFP (500/20, 515LP dichroic) excitation, a beam splitter (DualView, Photometrics) with a 505LP dichroic mirror,

emission filters for CFP (480/30) and YFP (535/40), and an electron-multiplied charge coupled device camera (Evolve 512, Photometrics). CFP and YFP images upon CFP excitation were captured every 5 s with 100 ms illumination time. FRET was monitored in real-time with the MetaFluor 5.0 software (Molecular Devices) as the ratio between YFP and CFP emission. The YFP emission was corrected for direct excitation of YFP at 436 nm and the bleedthrough of CFP emission into the YFP channel as previously described (*Börner et al., 2011*).

Larval preparations expressing Epac1-camps in lch5 neurons (*iav-GAL4>UAS-Epac1-camps*) were imaged at RT and stimulated with FSK (0.5 or 1 µM) at the beginning of the experiment to accumulate cAMP and decrease the FRET signal to a plateau phase (low forskolin response). 0.5 µM and 1 µM FSK elicited the same amplitude of FRET changes and the results were pooled accordingly. The amplitude of the low forskolin response was calculated by averaging five data points immediately before the stimulation and at the plateau phase. The difference was expressed as a percentage of maximal FRET response, obtained by application of IBMX (100 µM) followed by additional forskolin stimulation (10 µM). Piezo-actuated stimulation was performed only during the plateau phase (10 sweeps of $3 \times 1$ s 900 Hz stimulation separated by 1 s rest, 1 s inter-sweep interval).

The amplitude of the piezo-induced FRET change was calculated by averaging five data points immediately before and at the end of the mechanical stimulation block. The difference was expressed as a percentage of the low FSK response. Two quality criteria were used to assess cell health and failure to meet these resulted in exclusion of samples from further analysis: (1) stimulation with low FSK concentrations produced a FRET change and (2) did not saturate the sensor (i.e. subsequent stimulation with 10 µM FSK and 100 µM IBMX further decreased the FRET signal).

## G protein coupling assays

### Peptide synthesis

Peptides were synthesized using standard Fmoc-chemistry on an automated peptide synthesizer MultiPep (Intavis AG). Final side chain deprotection and cleavage from the solid support was achieved using TFA, water and thioanisole (95:2.5:2.5 vol%). Peptides were subsequently purified to >95% purity by preparative RP-HPLC (Shimadzu LC-8) equipped with a $300 \times 25$ mm PLRP-S column (Agilent). For both analytical and preparative use, the mobile phases were water or acetonitrile, respectively, each containing 0.1% TFA. Samples were eluted with a linear gradient of 5–90% acetonitrile in water: 30 min for analytical runs and 90 min for preparative runs. Peptide characterization by analytical HPLC (Agilent 1100) and MALDI-MS (Bruker Microflex) yielded the expected [M+H]+ mass peaks. Peptides were dissolved in DMSO to 100 mM and stored at 4° C until use.

### In vitro expression analysis and functional assays

For expression analyses and functional assays, transiently transfected COS-7 cells were used. COS-7 cells were cultivated in Dulbecco's Modified Eagle Medium (DMEM) supplemented with 10% fetal bovine serum, 100 U/ml penicillin and 100 µg/ml streptomycin at 37°C and 5% $CO_2$ in a humidified atmosphere. For enzyme-linked immunosorbent assays (ELISA) to determine cell surface expression, cells were split into 48-well plates ($3.8 \times 10^4$ cells/well), for total ELISA into 6-well plates ($3 \times 10^5$ cells/well) and for cAMP accumulation or IP assays into 96-well plates ($2 \times 10^4$ cells/well). After 24 hr cells were transfected with 0.5 µg/well receptor-encoding plasmid DNA for detecting cell surface expression, 1 µg/well for detecting total expression and 0.2 µg/well for analyzing response to peptides in functional assays using Lipofectamine 2000 (Invitrogen) according to manufacturer's protocol.

For an estimation of total and cell surface expression, receptors carrying an N-terminal HA were analyzed with a rat anti-HA-peroxidase antibody (Roche) in indirect cellular ELISA as described previously (*Schöneberg et al., 1998*).

To determine cAMP accumulation, COS-7 cells were washed 48 hr post transfection for 5 min with serum- and phenol red-free DMEM containing 1 mM IBMX. For analysis of agonistic peptides transfected cells were treated with 1 mM peptide in this cell medium.

Incubation was stopped by aspirating medium and lysing cells in LI buffer (PerkinElmer Life Sciences). Samples were frozen at −20°C and thawed for detection of cAMP concentrations using the AlphaScreen cAMP assay kit (PerkinElmer Life Sciences) according to manufacturer's protocol and the Fusion AlphaScreen multilabel reader (PerkinElmer Life Sciences).

For IP accumulation assays, the IP-One HTRF assay kit (CisBio) was used according to manufacturer´s protocol. In brief, transfected COS-7 cells were washed 48 hr post transfection with PBS and subsequently stimulated with 1 mM peptide in stimulation buffer (CisBio) for 30 min at 37°C. Incubation was terminated by lysing cells in lysis buffer on ice for 10 min and subsequent freezing at −20°C. Cell lysates were defrosted and subject to IP measurements in a 384-well format using the EnVision multilabel reader (PerkinElmer Life Sciences).

## Statistics

Data were analyzed in Prism 5.0 (GraphPad). Group means were compared by two-tailed Student's *t*-test. Where the assumption of normality of the sample distribution was violated as indicated by the D'Agostino and Pearsons omnibus normality test, group means were compared by two-tailed Mann-Whitney U test. Where indicated in figures asterisks denote the level of significance: $*p \leq 0.05$, $**p \leq 0.01$, $***p \leq 0.001$.

## Acknowledgements

We thank E Jöst and R Gueta for their help with the optogenetic actuators, D Bunsen and F Helmprobst for assistance with scanning electron microscopy, N Ehmann for discussions, and M Lohse and M Heckmann for support. This work was supported by grants from the Deutsche Forschungsgemeinschaft to TL (FOR 2149/P01 and P03, SFB 1047/A05, TRR 166/C03, LA2861/7-1), RJK (FOR 2149/P03, SFB 1047/A05, TRR 166/B04, KI1460/4-1), GN (SFB 1047/A03), MS (TRR 166/A04 and B04) and SP (FOR 2149/P02). GN acknowledges support provided through the Prix-Louis-Jeantet. Stocks obtained from the Bloomington *Drosophila* Stock Center (NIH P40OD018537) were used in this study.

## Additional information

### Funding

| Funder | Grant reference number | Author |
|---|---|---|
| Deutsche Forschungsgemeinschaft | FOR 2149/P01 | Tobias Langenhan |
| Deutsche Forschungsgemeinschaft | SFB 1047/A05 | Tobias Langenhan<br>Robert J Kittel |
| Deutsche Forschungsgemeinschaft | FOR 2149/P03 | Tobias Langenhan<br>Robert J Kittel |
| Deutsche Forschungsgemeinschaft | TRR 166/C03 | Tobias Langenhan |
| Deutsche Forschungsgemeinschaft | LA2861/7-1 | Tobias Langenhan |
| Deutsche Forschungsgemeinschaft | TRR 166/B04 | Markus Sauer<br>Robert J Kittel |
| Deutsche Forschungsgemeinschaft | KI1460/4-1 | Robert J Kittel |
| Deutsche Forschungsgemeinschaft | SFB 1047/A03 | Georg Nagel |
| Deutsche Forschungsgemeinschaft | TRR 166/A04 | Markus Sauer |
| Deutsche Forschungsgemeinschaft | FOR 2149/P02 | Simone Prömel |

The funders had no role in study design, data collection and interpretation, or the decision to submit the work for publication.

## Author contributions

NS, MN, Investigation, Visualization, Methodology, Writing—review and editing; CG, AG, IM, SG, SB, Formal analysis, Investigation, Visualization, Methodology, Writing—review and editing; MP, Investigation, Methodology, Writing—review and editing; MS, Resources, Funding acquisition, Investigation, Visualization, Writing—review and editing; EA, Resources, Investigation, Visualization, Writing—review and editing; SR, Resources, Methodology, Generation and bench-marking of peptides; JW, Formal analysis, Investigation, Methodology; SP, Formal analysis, Funding acquisition, Investigation, Methodology; GN, Resources, Formal analysis, Funding acquisition, Investigation, Visualization, Writing—review and editing; TL, RJK, Conceptualization, Resources, Formal analysis, Supervision, Funding acquisition, Investigation, Visualization, Methodology, Writing—original draft, Project administration, Writing—review and editing

## Author ORCIDs

Tobias Langenhan, http://orcid.org/0000-0002-9061-3809
Robert J Kittel, http://orcid.org/0000-0002-9199-4826

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
