## [Decision Letter]

Thank you for submitting your article "Mechano-dependent signaling by Latrophilin/CIRL quenches cAMP in proprioceptive neurons" for consideration by *eLife*. Your article has been reviewed by two peer reviewers, and the evaluation has been overseen by Richard Aldrich as the Senior Editor and Hugo Bellen as the Reviewing Editor. The following individuals involved in review of your submission have agreed to reveal their identity: Wayne Johnson (Reviewer #1); Daniel F Eberl (Reviewer #2). Both reviewers are very positive and we commend you for your fine paper. Please address the comments of the reviewers which are all minor.

The reviewers have discussed the reviews with one another and the Reviewing Editor has drafted this decision to help you prepare a revised submission.

General assessment and major comments:

Reviewer 1:

This is an impressive and well written manuscript with an abundance of data supporting the authors conclusions that the dCirl protein in larval chordotonal neurons plays a role in mechanosensory signaling. This manuscript is an extension of earlier work from the authors linking dCirl to mechanosensation using genetic, behavioral, electrophysiological and imaging approaches. Previous work characterized a dCirl mutant strain demonstrating dCirl-dependent mechanosensory signaling in chordotonal organs. Key findings provided in this manuscript attempt to further describe the role of aGPCR structural domains in neuron-specific function and to identify downstream metabotropic signaling pathways modified by dCirl activation. The bulk of the work is clearly presented and interpreted justifiably to support major conclusions of the studies. These results make a significant contribution to the area and are deserving of publication.

*Reviewer 2:*

The manuscript titled 'Mechano-dependent signaling by Latrophilin/CIRL quenches cAMP in proprioceptive neurons' by Scholz et al. is an exciting new paper which for the first time provides a molecular model for signal transduction for adhesion GPCRs in *Drosophila*. The authors show that Latrophilin/dCIRL, which is a *Drosophila* adhesion GPCR, is essential for both the initial step of mechanreception (conversion of mechanical stimuli to receptor current) and also downstream, when the second messenger cascade is regulated through decrease of cAMP in chordotonal neurons in larval lch5. Furthermore, this activity does not affect general excitability of the neurons, since thermosensory function and ontogenetic activation of the chordotonal neurons are not affected by dCIRL-KO. This paper is a continuation of the first paper by Scholz et al., 2015, where they show that dCIRL is involved in the initial mechanotransduction step in *Drosophila* proprioception and that it genetically interacts with the mechanically gated NompC channel. Here they characterize with more precision the subcellular localization within the cilia and neuronal dendrites of the chordotonal organs, localizing adjacent to TRPN1/ NompC which is the site of ionotrophic mechanosensation, shedding light on the molecular function of the GPCR. This is the first direct evidence showing how the *Drosophila* aGPCR receptor dCIRL/Latrophilin modulates the transduction of mechanical stimuli to electrical signal (receptor current) in the chordotonal neurons in larval lch5. The authors also show that the length of extracellular N-terminal fragment (NTF) of dCIRL and the tethered agonist of the receptor (Stachel) is critically important for mechanotransduction. They further demonstrate that this process is independent of the autoproteolytic property of dCIRL through the GAIN domain. This study has been conducted carefully and is technically sound. This paper is important in understanding the molecular cascade from the first initiation of mechanical signal to how the signal is transmitted further downstream to specific intracellular destinations in the sensory neuron.

Minor comments:

Experiments examining the effect of varied lengths of dCirl NTF on response frequency (Figure 4) are perhaps overstated. It cannot be concluded from these few experiments whether the NTF inserts directly affect interactions with extracellular matrix components or simply destabilize the protein leading to some loss of function. Since "expression" of the dCirl[3XCD4] variant results in essentially a null phenotype, it is not clear whether the variant protein is inherently unstable or is not expressed at all leading to the loss-of-function phenotype. I could not find evidence of stable expression of the dCirl[3XCD4] variant in vivo in the manuscript. Demonstration of expression using Western blot or imaging is used for numerous other modified constructs in the manuscript but I could not find the data for this one protein. If it is there somewhere please indicate. Otherwise, please provide evidence of stable expression. In addition, the concluding sentence of that section: "We conclude that ECD length and tensile properties adjust dCIRL's response towards mechanical challenge" is vastly overstated based upon this minimal set of constructs and experiments. This concluding sentence should be modified to more accurately reflect the significance of these experiments. Although I am not requesting the removal of this entire section from the manuscript, it does not add significantly to the manuscript without a more extensive characterization that would be outside the scope of this paper.

A significant component of the model for the role of dCirl in mechanosensory neurons is the functional relationship with previously characterized mechanosensory ionotropic channels such as the TRP channels nompC and nanchung. Previously published work from the authors demonstrated clear genetic interactions between the dCirl mutant and both nompC and nan mutants. Figure 1 provides images demonstrating the subcellular localization of dCIRL in dendrites and cilium along with nompC localization. In the figure legend, it is stated that dCirl is subcellularly localized "next to the TRP channel NompC". This statement suggests a subcellular colocalization of dCirl and nompC consistent with some sort of direct interaction or adjacent signaling mechanism. Since the spatial relationship between dCirl and nompC proteins in the membrane is not known and a direct physical contact is not essential to the proposed model, this is a confusing statement. Although still valid, the data supports only a possible coexpression in similar subcellular regions of the chordotonal neurons. The statement that dCirl is localized "next to" nompC is imprecise and overstated. This should be rephrased to more accurately reflect the strength and significance of the results.

1) Figure 2: In 2B – please clarify what does the n represent – number of oocytes recorded from? In 2C it is not clear which bar is significant in relation to other bars on the bar graph. Please use brackets to denote the statistical significance between the two bars. In 2G the figure legend in says 'number denotes p values of comparison of event frequency' however it seems that the p value is not given in the figure.

2) Figure 3: In 3B, mechanosensory signal has been recorded from the lch5 for n=8. It is not clear whether these are recording from 8 individual animals or 8 different lch5 organs (in fewer animals or even a single animal). Please clarify.

3) Same comment regarding Figure 4, where the lch5 is recorded for effect of elongation of the NTF. The authors need to make clear the number of animals the data represent.

4) ChR2XXM and Chop2XXM have been used interchangeably in figure and main body of text (Figure 2 expanded genotype explained in the subsections “Fly culture conditions and stocks” and “Chordotonal neurons”). Please use only one denotation, perhaps at first mention indicate the alternate terminology, but stick to only one throughout.

5) Figure 6: In 6D it has been very nicely shown that pharmacological inhibition of adenylyl cyclase by SQ22536 can rescue mechanically evoked current in the dCIRLKO. Perhaps it would be more conclusively demonstrated if a negative control such as a NompC null mutant is also included showing that the pharmacological rescue of dCIRLKO mutant is dependent on mechanical stimulation? This is a relatively minor suggestion, and we do not feel strongly that this experiment must be done before publication.

6) In Figure 1 – please indicate on the bar graph which one is the control and the dCIRLKO.

7) Discussion, fifth paragraph: spelling of matricellular.

8) Discussion, fifth paragraph: data support (not data supports).

9) Discussion, eighth paragraph: spelling of cyclase.

10) Discussion, last paragraph: aGPCR (not aGPCRs).

---

## [Author Response]

*Minor comments:*

*Experiments examining the effect of varied lengths of dCirl NTF on response frequency (Figure 4) are perhaps overstated. It cannot be concluded from these few experiments whether the NTF inserts directly affect interactions with extracellular matrix components or simply destabilize the protein leading to some loss of function. Since "expression" of the dCirl[3XCD4] variant results in essentially a null phenotype, it is not clear whether the variant protein is inherently unstable or is not expressed at all leading to the loss-of-function phenotype. I could not find evidence of stable expression of the dCirl[3XCD4] variant* in vivo *in the manuscript. Demonstration of expression using Western blot or imaging is used for numerous other modified constructs in the manuscript but I could not find the data for this one protein. If it is there somewhere please indicate. Otherwise, please provide evidence of stable expression.*

We now provide evidence for stable expression of the *dCirl[3XCD4]* variant allele product by Western blot against poly-V5 tags inserted into each of the CD4 modules. The blot was performed on protein extracts from animals expressing the dCirl[3XCD4] protein, and two negative controls lacking the 3xCD4 spacers were included for specificity control. Full length protein has a calculated molecular weight of appx. 298 kDa, NTF-only dCirl[3XCD4] of appx. 177 kDa. As the GPS of this allele is cleavable only the NTF containing the CD/V5 modules is visible on the SDS gel after denaturing.

Western blot analysis shows the stable expression of the dCIRL^3xCD4^ fusion protein in vivo. No full-length protein is visible indicating that autoproteolytic cleavage is not affected by the length extension of the NTF.

We have documented these results in a new panel in Figure 4 (now Figure 4), inserted references to it at the appropriate place within the manuscript (subsections “dCIRL NTF length determines mechanosensitivity of chordotonal neurons” and) and extended the related methods section by the western blot protocol (subsection “Immunoblots”).

*In addition, the concluding sentence of that section: "We conclude that ECD length and tensile properties adjust dCIRL's response towards mechanical challenge" is vastly overstated based upon this minimal set of constructs and experiments. This concluding sentence should be modified to more accurately reflect the significance of these experiments. Although I am not requesting the removal of this entire section from the manuscript, it does not add significantly to the manuscript without a more extensive characterization that would be outside the scope of this paper.*

We changed the sentence to reflect the results with more caution: “We therefore hypothesize that ECD length and tensile properties may adjust dCIRL’s response towards mechanical challenge (Figure 4).”

*A significant component of the model for the role of dCirl in mechanosensory neurons is the functional relationship with previously characterized mechanosensory ionotropic channels such as the TRP channels nompC and nanchung. Previously published work from the authors demonstrated clear genetic interactions between the dCirl mutant and both nompC and nan mutants. Figure 1 provides images demonstrating the subcellular localization of dCIRL in dendrites and cilium along with nompC localization. In the figure legend, it is stated that dCirl is subcellularly localized "next to the TRP channel NompC". This statement suggests a subcellular colocalization of dCirl and nompC consistent with some sort of direct interaction or adjacent signaling mechanism. Since the spatial relationship between dCirl and nompC proteins in the membrane is not known and a direct physical contact is not essential to the proposed model, this is a confusing statement. Although still valid, the data supports only a possible coexpression in similar subcellular regions of the chordotonal neurons. The statement that dCirl is localized "next to" nompC is imprecise and overstated. This should be rephrased to more accurately reflect the strength and significance of the results.*

We thank the reviewer for the comment and have adapted the wording according to his suggestion. The sentence in caption for Figure 1: “SIM imaging shows *dCirl^N-RFP^* (magenta) in the distal dendrite (arrowheads) extending to the ciliary compartment, where the receptor is coexpressed in the same subcellular region with the TRP channel NompC (green).”

*1) Figure 2: In 2B please clarify what does the n represent – number of oocytes recorded from? In 2C it is not clear which bar is significant in relation to other bars on the bar graph. Please use brackets to denote the statistical significance between the two bars. In 2G the figure legend in says 'number denotes p values of comparison of event frequency' however it seems that the p value is not given in the figure.*

Figure 2 represents the number of individual oocytes recorded from. We have added a statement to the figure caption to clarify this.

Figure 2: We have modified Figure 2 by inserting brackets to indicate the relationship of statistically compared datasets.

Figure 2: We apologize for omitting the p values as indicated in the figure caption. We have now inserted them in the figure panel.

*2) Figure 3: In 3B, mechanosensory signal has been recorded from the lch5 for n=8. It is not clear whether these are recording from 8 individual animals or 8 different lch5 organs (in fewer animals or even a single animal). Please clarify.*

In this experiment, n=8 denotes lch5 recordings from 8 different animals. We have added a statement to the figure caption to clarify this.

*3) Same comment regarding Figure 4, where the lch5 is recorded for effect of elongation of the NTF. The authors need to make clear the number of animals the data represent.*

As in all other experiments, n equals lch5 recordings from individual animals. We have added a statement to the figure caption to clarify this (Figure 4 legend).

*4) ChR2XXM and Chop2XXM have been used interchangeably in figure and main body of text (Figure 2 expanded genotype explained in the subsections “Fly culture conditions and stocks” and Chordotonal neurons”). Please use only one denotation, perhaps at first mention indicate the alternate terminology, but stick to only one throughout.*

We respectfully note that both terms are not used interchangeably:

Chop2 denotes the apoprotein (without retinal), which we have genetically encoded and transgenically expressed in the animals as described. Therefore, all references indicating genotypes use Chop2-XXM (=channelopsin) as descriptor, e.g. *UAS-chop2^XXM^*.

ChR2-XXM (=Channelrhodopsin) denotes the apoprotein that is bound to its chromophore retinal after protein translation. ChR2 therefore is the photoactivatable version of Chop2 that we used in functional assays. All references to the functional interrogation of or with the optogenetic actuator therefore use ChR2-XXM as descriptor.

We feel that for clarity and also to be in accordance with terminological conventions in the optogenetics field the distinction between both terms should be maintained. We also fully agree that a clear denotation should be used and have now briefly defined both terms at first mention (subsection “Optogenetic stimulation of chordotonal neurons bypasses dCIRL-dependence”, first paragraph).

*5) Figure 6: In 6D it has been very nicely shown that pharmacological inhibition of adenylyl cyclase by SQ22536 can rescue mechanically evoked current in the dCIRLKO. Perhaps it would be more conclusively demonstrated if a negative control such as a NompC null mutant is also included showing that the pharmacological rescue of dCIRLKO mutant is dependent on mechanical stimulation? This is a relatively minor suggestion, and we do not feel strongly that this experiment must be done before publication.*

We thank the reviewer for the comment and indeed such an experiment would further strengthen the conclusions of this part of the manuscript. We will conduct such control measurements in due course but, as the reviewer suggests, would leave the dataset in the current manuscript as is.

*6) In Figure 1 – please indicate on the bar graph which one is the control and the dCIRLKO.*

We apologize for the oversight and have added the genotype designations to Figure 7 and Figure 7—figure supplement 1.

*7) Discussion, fifth paragraph: spelling of matricellular.*

Corrected.

*8) Discussion, fifth paragraph: data support (not data supports).*

Corrected.

*9) Discussion, eighth paragraph: spelling of cyclase.*

Corrected.